# The Role of Family Physicians in a Pandemic: A Blueprint

**DOI:** 10.3390/healthcare8030198

**Published:** 2020-07-05

**Authors:** Jie Qi Lee, Wayren Loke, Qin Xiang Ng

**Affiliations:** 1NTU Lee Kong Chian School of Medicine, 11 Mandalay Road, Singapore 308232, Singapore; leej0150@e.ntu.edu.sg; 2MOH Holdings Pte Ltd., 1 Maritime Square, Singapore 099253, Singapore; wayren.loke@mohh.com.sg

**Keywords:** primary care, general practitioners, pandemic, readiness, blueprint

## Abstract

Pandemics are a significant stress test for a country’s economic, political and health systems. An effective pandemic response demands a multi-pronged and multi-layered approach, comprising surveillance, containment, border control, as well as various social and community measures. In the wake of the novel coronavirus disease 2019 (COVID-19) pandemic, which has now infected more than 7 million people worldwide, strict quarantine measures are a commonplace, and a third of the world’s population have now gone into some form of lockdown. With the exception of border control, all these response measures involve the contributions of family physicians and general practitioners (GPs) in one way or another. Primary care physicians form and lead the primary care network, which in turn forms the backbone of any healthcare system. Being the first point of contact for a significant proportion of patients, primary care physicians play an essential strategic function in the fight against disease, both during peacetime and in the event of a public health crisis. In this commentary, we examine and propose some of the key roles that they play in a pandemic, drawing examples from the current COVID-19 pandemic and past experiences. COVID-19 has showed us that the world is grossly unprepared for a pandemic, both in terms of our global management and the structure of our current primary health care systems, and this should provide the impetus for us to improve.

## 1. Introduction

Pandemics are a significant stress test for a country’s economic, political and health systems. An effective pandemic response demands a multi-pronged and multi-layered approach comprising surveillance, containment, border control, as well as various social and community measures. With the exception of border control, all these response measures involve the contributions of family physicians and general practitioners (GPs) in one way or another. This comes as no surprise, as they form and lead the primary care network, which in turn forms the backbone of any healthcare system. Being the first point of contact for a significant proportion of patients, primary care physicians play an essential strategic function in the fight against disease, both during peacetime and in the event of a public health crisis.

In this commentary, we examine and propose some of the key roles they play in a pandemic, drawing examples from the current novel coronavirus disease 2019 (COVID-19) pandemic and past experiences. These pertinent decisions and services include the (1) triage and treatment of suspected or confirmed cases, (2) resource allocation, (3) surveillance and monitoring, (4) preventive care, (5) provision of affordable care and (6) ongoing delivery of primary care to patients with other acute illnesses and chronic diseases.

## 2. Triage and Treatment

In the wake of the novel coronavirus disease 2019 (COVID-19) pandemic, which has now infected more than 7 million people worldwide, strict quarantine measures are a commonplace, and a third of the world’s population have now gone into some form of lockdown [1]. Primary care physicians form the first line of defense in any healthcare system. Patients typically present first to their family physicians or GPs, who then makes a judgement call if the ailment can be addressed then and there, or if escalation of care is required. A robust primary care network of family physicians and GPs therefore serves as the bedrock of an efficient healthcare system. It decreases the burden on secondary and tertiary facilities, thereby reducing healthcare costs for the country [2]. This blueprint applies to almost all medical scenarios, and a pandemic situation is no exception.

During the H1N1-2009 pandemic, as well as the ongoing COVID-19 pandemic, clinics across Singapore have implemented strict criteria-based screening, referring only patients fulfilling case definitions to the emergency departments or the National Centre for Infectious Diseases (NCID) [3]. As local transmission became more prevalent, primary care clinics were subsequently enlisted to perform COVID-19 testing for a select subgroup of low-risk (medically stable) patients, under the “Swab and Send Home” program [4]. Without such community treatment capacities and a gatekeeping system, numerous cases of the common cold and even allergies with upper airway symptoms would have made their way to the already overwhelmed emergency departments of Singapore. This may be especially relevant to the COVID-19 pandemic, as the majority of patients are medically stable and have mild to no symptoms [5].

In the case of the H1N1-2009 pandemic, GPs also played the additional role of treating patients with Oseltamivir, to reduce their likelihood of developing complications requiring hospitalization, freeing up resources for severe and critical cases [6].

## 3. Resource Allocation

As resources are finite, pandemic situations often result in resource scarcity, with anti-viral drugs, vaccines, ventilators and personal protective equipment (PPE) becoming in short supply. At the national level, health ministries would have performed one round of resource allocation as a safety net. In Singapore, resources are apportioned in a tiered manner. The Ministry of Health (MOH) first allocates resources to healthcare workers in public institutions, followed by Public Health Preparedness Clinics (PHPCs), which provide subsidized treatment to patients with respiratory symptoms, and subsequently to all other primary care clinics [7]. This is based on the principle that resources should be given to those most ill, or who are integral to a functioning society during a crisis [8]. This does not, however, absolve the need for rationing at the GP level. The GP would still need to make calculated decisions on which patients would benefit most from these curative or preventive therapies; failing this would undermine the abovementioned sieve function of primary care and upset the balance laid out at the national level. The Oseltamivir shortage during the H1N1-2009 pandemic is a case in point. During the early stages of the pandemic, the indiscriminate prescription of the anti-viral drug by pharmacies and clinics resulted in them ending up in the undeserving hands of patients with mild disease, and even hoarders in some countries, threatening the ability of national stockpiles to cater to high-risk and critically ill patients later on [9].

## 4. Surveillance and Monitoring

Perhaps more relevant to a pre-pandemic situation, primary care physicians have a huge potential to take on a surveillance and monitoring role. Being the first point of contact for many infectious diseases, a network of sentinel primary care physicians could potentially serve as an early warning system for detecting sudden spikes in disease incidences or the emergence of novel strains. Primary care clinics in the US, for instance, form a large component of the US Outpatient Influenza-like Illness Surveillance Network (ILINet), where weekly percentages of patient visits for ILI are monitored and compared against the national baseline [10]. In Singapore, doctors at the Sims Drive clinic helped to uncover the 2006 Zika outbreak in Singapore, after witnessing an unusual increase in patients presenting with dengue-like symptoms, and flagging these cases up to the Singapore Ministry of Health (MOH) [11]. Whilst not an example of a pandemic situation, it would not be difficult to imagine such an alarm system being extrapolated to a pandemic situation.

The utility of primary care as an early-warning system, however, remains limited in many countries, due to the lack of a strongly interconnected healthcare system [12], and reservations from both doctors and patients regarding the sharing of electronic medical records, even with the relevant authorities and public health officials [13].

In Singapore, the government has set up the National Electronic Health Record (NEHR), which is a secure system that collects summary electronic patient medical records across different healthcare providers. However, patients can choose to opt out of the NEHR and data contribution to the NEHR is voluntary for all private healthcare providers [14].

## 5. Preventive Care

Offering preventive services and opportunistic health promotion is part of a primary care physician’s DNA. Advocating for age-appropriate cancer screening and vaccinations against preventable diseases is part and parcel of their daily practice. The latter is particularly pertinent in a pandemic situation, for various reasons. Most obviously, vaccines can protect the population against the culprit bug. For instance, an estimated 300 deaths and 1 million illnesses were prevented by the pandemic H1N1 vaccine in the US [15]. Even if vaccines against the offending pathogen are unavailable, they still serve a few purposes. Influenza coverage during the ongoing COVID-19 pandemic, for one, guards against flu complications requiring hospitalization. After all, the last thing that an overstretched healthcare system needs during this time is a simultaneous influenza outbreak. Influenza prevention may make COVID-19 detection more efficient as well. Influenza and COVID-19 are virtually clinically indistinguishable, and any attempt at reducing the haystack of respiratory presentations would theoretically be of great help, especially in countries experiencing test kit shortages [16]. Lastly, influenza vaccinations could reduce the risk of coinfections, the outcome of which is still being studied [17].

While a physician’s waiting room may be the last place anyone wants to be during a pandemic, GPs can provide opportunistic vaccinations when patients come to consult for separate issues, e.g., a urinary tract infection. In Australia, GPs have also gone the extra mile to launch drive-through flu vaccination services, providing influenza vaccinations to patients from the comfort and safety (distance) of their own car [18].

Patient education is also a key tenet of primary care medicine. Owing to their enduring, trusting relationships with patients, GPs are uniquely suited to provide much needed clarity and solutions, especially during the disorienting times of a pandemic. To illustrate, a survey conducted by the College of Family Physicians of Canada found that an overwhelming majority (86%) of respondents expressed that they should be able to turn to their family doctor for information and advice at a time of a serious medical emergency [19]. In a rapidly evolving pandemic situation, it is easy for the layperson to get lost in a sea of information, and not uncommonly, misinformation that is unhelpful at best, and fatal at worst. This may be especially true for the ongoing COVID-19 pandemic, as there is still much uncertainty and speculation surrounding the epidemiology and virology of COVID-19, including the strong possibility of asymptomatic spreaders and the incubation period for clinical disease [20].

In an extreme example, an Arizonian man died from consuming a form of chloroquine used to clean aquariums, days after the anti-malarial drug was touted by some as a potential cure for COVID-19 [21]. Armed with clinical expertise and evidence-based medicine, primary care physicians can not only reinforce and clarify the basis behind public messages promulgated by authorities, but also dispel myths circulating through social media and hearsay. This is especially vital for marginalized communities who may not have access to credible, comprehensible sources of information [7]. Healthcare inequalities are often further exposed in a pandemic situation [22]. During the H1N1-2019 pandemic, for instance, family doctors in the US serving migrant and seasonal farm worker communities transmitted Centre for Disease Control and Prevention (CDC) public health messages and bilingual patient education tools directly to patients, as well as through regional migrant health coordinators [11].

With insight into each patient’s unique health and life circumstances, primary care physicians are well-positioned to provide personalized recommendations that public announcements are unable to. They can also reiterate to patients the importance of good personal hygiene and social distancing to stem the spread of COVID-19.

## 6. Provision of Affordable Care

Pandemics often result in trying times for the economy. In the last week of March 2020 alone, more than six and a half million people filed for unemployment benefits in the US, due to the ramifications of the COVID-19 pandemic [23]. It would be extremely detrimental to disease containment efforts if patients were denied access to testing and treatment because of financial constraints. In Singapore, GPs help to mitigate this problem by participating in the PHPC scheme [24]. Under the scheme, citizens and permanent residents enjoy highly subsidized and even free consultations and treatment when they visit a participating clinic for respiratory complaints. Additionally, ongoing community health assist scheme (CHAS) and public assistance (PA) subsidies ensure that low-income patients do not fall through the cracks, and they continue to have equitable access to chronic care, even through tough circumstances.

There are also increasing reports of high rates of anxiety, insomnia and depressive symptoms amongst the general public and frontline medical staff, especially during a lockdown [25,26]. An effective pandemic response must also include a mental health response, and GPs could be an effective part of this response. Healthcare cost should not be a prohibitive factor for people with emotional anguish and serious mental health issues.

## 7. Continual Provision of Essential Services

Last but not least, businesses and operations may come to a halt during a lockdown but diseases and exacerbations of chronic diseases do not. In addition to their aforementioned roles, family physicians provide the ongoing delivery of primary care to patients with other acute illnesses and chronic conditions. During the national lockdown for COVID-19, Italy reported significant decreases (ranging from 73% to 88%) in pediatric emergency department visits [27], as well as a 48.4% reduction in admissions for acute myocardial infarction (AMI) [28]. This was accompanied by reports of delayed access to hospital care, with a corresponding increase in fatality and complication rates [27,28]. These are concerning trends, and they signal the need for community care to continue, even during a pandemic situation.

Any major outbreak would also have detrimental effects on the mental health of individuals and society, as we have experienced during the 2003 SARS outbreak and now with SARS-CoV-2 [25]. During a pandemic, however, family physicians have the added responsibility of preventing inter-patient transmission within their premises, especially when comorbidities often confer poorer clinical outcomes in an infection [29]. They do so through a variety of approaches, with adequate segregation, sanitization and physical distancing being the most basic of them. To bypass physical contact altogether, some family physicians leverage on telemedicine and medication delivery services. In recognition of its potential to relieve hospital workloads and interrupt transmission during COVID-19, the South Korean Ministry of Health and Welfare temporarily lifted its ban on telemedicine, allowing numerous vulnerable patients to have access to continuing primary care from the safety of their homes [30]. The American Academy of Family Physicians (AAFP) also supports the expanded use of telemedicine as an important and efficient tool to help care for patients as the COVID-19 pandemic rages on [31]. As resources could be particularly scarce during a serious pandemic situation, timely psychological support could also take many forms [32]. The utilization rates of remote services, however, remain low globally consequent upon legal uncertainty, payment issues and technical challenges, as well as patient and physician unfamiliarity [31,33].

An engaged multidisciplinary team is also critical when it comes to health promotion and chronic disease management. GPs should work with allied health professionals and community care teams for the effective coordination and continuity of multidisciplinary care. A collaborative working relationship between GPs and allied health professionals is desirable and essential for a strong primary care network, but this is often difficult to achieve, as previous studies have revealed [34,35].

## 8. Conclusions

Primary care capabilities, and hence, primary care physicians, play pivotal roles in a pandemic situation. They aid in the triage and treatment process, help educate patients, and render affordable care and preventive care (as illustrated in Figure 1). As Dr Victor Johnston, renowned family physician and founder of the College of General Practitioners of Canada, once said,
General practitioners … are the doctors closest to [the] people. They heal more of the broken-hearted, repair more of the injured and deprived, and live with the poor and dying who are without influence and hope. [36]

COVID-19 is not the first pandemic that the world has seen and, with ever increasing levels of travel, global interconnectedness and cross-border movement, it definitely will not be the last. Family physicians need to respond to the increasing need of people and nations. The quiet, conscientious courage displayed by primary care physicians during a pandemic indubitably aids a nation’s response and recovery. Yet, a substantial amount of potential remains untapped, especially in the areas of surveillance, telemedicine and multidisciplinary community care teams. Perhaps COVID-19 will give the impetus for further development in these areas.

## Figures and Tables

**Figure 1 healthcare-08-00198-f001:**
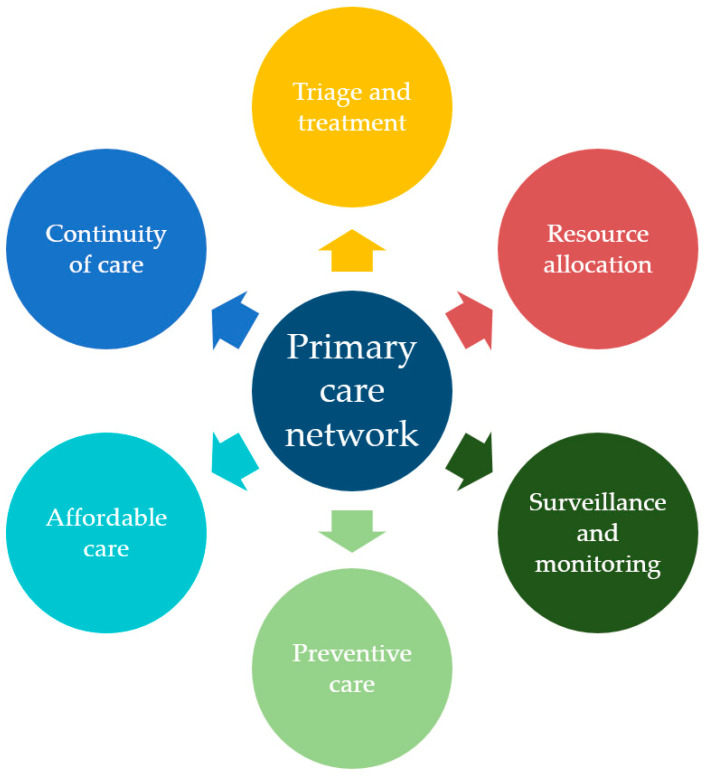
An illustration of the multifaceted role of the primary care network in which general practitioners (GPs) or family physicians operate.

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
