# Peer review of "The Role of Family Physicians in a Pandemic: A Blueprint"

_healthcare, 2020, doi:10.3390/healthcare8030198_

Round 1

Reviewer 1 Report

The particular aspect of Covid-19 is presented clearly and concisely.

Manuscript is dealing a particular aspect of Covid-19, but is well organized and very easy to follow.

There are some minor grammatical and spelling corrections in the text, that required to be rectified before publication.

Author Response

Thank you for the positive comments. We have done a close edit of the manuscript for grammatical and spelling errors.

Reviewer 2 Report

The submitted manuscript summarizes the major sectors where family physicians contribute, and may contribute during a pandemic, while underscoring their active role during the recent pandemic.

The manuscript is well written. There is a need to check for minor format, and spelling errors, including headlines and literature.

The content is well-organized and concrete. I would advise though an extension of the introductory part to facilitate the presentation of the following sub-topics. It also sounds like the abstract.

Chapters 5 & 7 could be merged or enriched.

I understand that the rationale of this work is to follow a timeline of pertinent decisions and services, and this should be described early in the text.

In addition, I believe that although the previous viral ourbreaks haven't reached today's proportions, it would be necesssary to include more examples and citations, preferably from peer-reviewed sources, even from restricted geographical areas.

As I mentioned above, the abstract and introductory part could be updated. The last sentence shows an "unpreparedness", and it is not clear whether it pertains to the global/social management or to the structure of the primary health care system. The following paragraphs provide examples of preparedness and remind us of what we could do. However, there have been many challenges for caregivers, and beneficiaries. Both sides are still hesitant to interact, and there have been several cases of fatal negligence. On the opposite site, because of social distancing, many communicable diseases have been reported low, especially in children. It would be useful to cite such examples, and provide us with some workload and perfomance metrics, and the reasons within.

Author Response

Comment 1: The content is well-organized and concrete. I would advise though an extension of the introductory part to facilitate the presentation of the following sub-topics. It also sounds like the abstract.

Reply 1: Thank you for the comment. We have further extended the introduction section as per your suggestion, "In this commentary, we examine and propose some of the key roles they play in a pandemic, drawing examples from the current COVID-19 pandemic and past experiences. These pertinent decisions and services include the (1) triage and treatment of suspected or confirmed cases, (2) resource allocation, (3) surveillance and monitoring, (4) preventive care, (5) provision of affordable care and (6) ongoing delivery of primary care to patients with other acute illnesses and chronic diseases."

Comment 2: Chapters 5 & 7 could be merged or enriched.

Reply 2: Thank you for the comment. These two sections are now merged as per your suggestion.

Comment 3: I understand that the rationale of this work is to follow a timeline of pertinent decisions and services, and this should be described early in the text.

Reply 3: This has now been described early in the introduction section.

Comment 4: In addition, I believe that although the previous viral ourbreaks haven't reached today's proportions, it would be necesssary to include more examples and citations, preferably from peer-reviewed sources, even from restricted geographical areas.

Reply 4: We have added additional examples as per your suggestion, for example, "In the case of the H1N1-2009 pandemic, GPs also played the additional role of treating patients with Oseltamivir to reduce their likelihood of developing complications requiring hospitalization, freeing up resources for severe and critical cases [6]."

Comment 5: As I mentioned above, the abstract and introductory part could be updated. The last sentence shows an "unpreparedness", and it is not clear whether it pertains to the global/social management or to the structure of the primary health care system. The following paragraphs provide examples of preparedness and remind us of what we could do. However, there have been many challenges for caregivers, and beneficiaries. Both sides are still hesitant to interact, and there have been several cases of fatal negligence. On the opposite site, because of social distancing, many communicable diseases have been reported low, especially in children. It would be useful to cite such examples, and provide us with some workload and perfomance metrics, and the reasons within.

Reply 5: Thank you for the useful comment. We have added examples of specific challenges, which we believe stems from both global/social management and the current structure of the primary health care system.

"COVID-19 has showed us that the world is grossly unprepared for a pandemic, both in terms of our global management and the structure of our current primary health care systems, and this should provide the impetus for us to improve."

"The utility of primary care as an early-warning system, however, remains limited in many countries due to the lack of a strongly interconnected healthcare system [12] and reservations from both doctors and patients regarding the sharing of electronic medical records, even with the relevant authorities and public health officials [13].

In Singapore, the government has set up the National Electronic Health Record (NEHR), which is a secure system that collects summary electronic patient medical records across different healthcare providers. However, patients can choose to opt out of the NEHR and data contribution to the NEHR is voluntary for all private healthcare providers [14]."

"An engaged multidisciplinary team is also critical when it comes to health promotion and chronic disease management. GPs should work with allied health professionals and community care teams for effective coordination and continuity of multidisciplinary care. A collaborative working relationship between GPs and allied health professionals is desirable but oftentimes difficult to achieve as previous studies have revealed [27,28]."

"Yet, a substantial amount of potential remains untapped especially in the areas of surveillance, telemedicine and multidisciplinary community care teams. Perhaps COVID-19 will give the impetus for the further development in these areas."

Reviewer 3 Report

Dear authors,

First of all, I would like to congratulate you for the initiative to describe the important role of the medical team working in Primary Health Care at a time when public health faces one of the greatest challenges of the century.

I describe below some considerations that may contribute to the improvement of production:

  1. The article deals with priority topics when dealing with a pandemic, but very superficially, they could have gone deeper into the items treated;
  2. The role of the medical team in surveillance, monitoring, data recording, patient screening, guidance and conducting interventions is very important, as described in the article, but the authors could address how the system works in Singapore and point out possible flaws and possibilities of solutions;
  3. The doctor's role is important, but without an engaged multidisciplinary team, the doctor will have difficulty in public health when it comes to prevention and promotion beyond assistance and this was not mentioned in the article.

Author Response

Comment 1: The article deals with priority topics when dealing with a pandemic, but very superficially, they could have gone deeper into the items treated.

Reply 1: Thank you for the comments. We have added additional information as requested, and we hope we have provided more detail and depth to our discussion to the reviewer's satisfaction (changes in the manuscript are highlighted in yellow).

Comment 2: The role of the medical team in surveillance, monitoring, data recording, patient screening, guidance and conducting interventions is very important, as described in the article, but the authors could address how the system works in Singapore and point out possible flaws and possibilities of solutions.

Reply 2: Thank you for the comments. We have now added that, "The utility of primary care as an early-warning system, however, remains limited in many countries due to the lack of a strongly interconnected healthcare system [11] and reservations from both doctors and patients regarding the sharing of electronic medical records, even with the relevant authorities and public health officials [12].

In Singapore, the government has set up the National Electronic Health Record (NEHR), which is a secure system that collects summary electronic patient medical records across different healthcare providers. However, patients can choose to opt out of the NEHR and data contribution to the NEHR is voluntary for all private healthcare providers [13]."

References

[11] Rust G, Melbourne M, Truman BI, Daniels E, Fry-Johnson Y, Curtin T. Role of the primary care safety net in pandemic influenza. Am J Public Health. 2009;99 Suppl 2:S316-23.

[12] Powell J, Fitton R, Fitton C. Sharing electronic health records: the patient view. Journal of Innovation in Health Informatics. 2006;14(1):55-7.

[13] Integrated Health Information Systems. About the national electronic health record. Available from: https://www.ihis.com.sg/Latest_News/Media_Releases/Pages/About_the_National_Electronic_Health_Record.aspx [last accessed 28 June 2020].

Comment 3: The doctor's role is important, but without an engaged multidisciplinary team, the doctor will have difficulty in public health when it comes to prevention and promotion beyond assistance and this was not mentioned in the article.

Reply 3: Thank you for the comment. We agree with the reviewer and have now added that, "An engaged multidisciplinary team is also critical when it comes to health promotion and chronic disease management. GPs should work with allied health professionals and community care teams for effective coordination and continuity of multidisciplinary care. A collaborative working relationship between GPs and allied health professionals is desirable but oftentimes difficult to achieve as previous studies have revealed [29,30]."

References

[29] Grant A, Mackenzie L, Clemson L. How do general practitioners engage with allied health practitioners to prevent falls in older people? An exploratory qualitative study. Australasian journal on ageing. 2015 Sep;34(3):149-54.

[30] Harris MF, Chan BC, Daniel C, Wan Q, Zwar N, Davies GP. Development and early experience from an intervention to facilitate teamwork between general practices and allied health providers: the Team-link study. BMC Health Services Research. 2010 Dec 1;10(1):104.

Round 2

Reviewer 2 Report

The authors made several changes to the manuscript.

It would be desirable to have some recent data, not only to describe the possibilities, but the challenges of the system as well.

I will bypass the malpractice/immunity approach that has been extremely popular for both professionals and beneficiaries lately. Though it still merits mention.

It is always helpful to provide some concrete evidence, instead of general ideas, e.g. the implementation of technology.

The reference to fall risk [31,32] is a nice example of continuous/preventive care, but I think that the FFN made substantial progress in recent years.

I will add some links for the authors to review, and possibly to integrate as such, or as perspectives, adding other references:

https://theconversation.com/98-of-emergency-calls-for-strokes-are-made-by-someone-else-so-what-if-youre-alone-in-lockdown-138348

https://www.aafp.org/practice-management/health-it/telemedicine-telehealth.html

https://www.mafp.com/news/coronavirus-what-michigan-family-physicians-need-to-know

https://www.nejm.org/doi/full/10.1056/NEJMp2006376?query=TOC

https://news.gallup.com/poll/307640/americans-worry-doctor-visits-raise-covid-risk.aspx

https://www.reidhealth.org/press-releases/emergencies-cant-wait-on-covid-19

https://www.cdc.gov/coronavirus/2019-ncov/hcp/framework-non-COVID-care.html

https://www.cebm.net/covid-19/preventing-non-covid-19-hospital-admissions-during-a-pandemic-a-rapid-overview-of-the-evidence-for-high-value-medications/

https://www.healthychildren.org/English/tips-tools/ask-the-pediatrician/Pages/Is-it-OK-to-call-the-pediatrician-during-COVID-19-even-if-Im-not-sure-my-child-is-sick.aspx

https://apps.who.int/iris/bitstream/handle/10665/204463/9789241510219_eng.pdf?sequence=1

https://khn.org/news/pediatric-practices-struggle-to-adapt-and-survive-amid-covid-19/

https://www.ncbi.nlm.nih.gov/pmc/articles/PMC7274978/

Author Response

Comment 1: It would be desirable to have some recent data, not only to describe the possibilities, but the challenges of the system as well.

I will bypass the malpractice/immunity approach that has been extremely popular for both professionals and beneficiaries lately. Though it still merits mention.

It is always helpful to provide some concrete evidence, instead of general ideas, e.g. the implementation of technology.

Reply 1: Thank you for the useful comments and many helpful references provided. We have added some recent data and also described the challenges faced with concrete evidence: 

"Last but not least, businesses and operations may come to a halt during a lockdown, but diseases and exacerbations of chronic diseases do not. In addition to their aforementioned roles, family physicians provide ongoing delivery of primary care to patients with other acute illnesses and chronic conditions. During the national lockdown for COVID-19, Italy reported significant decreases (ranging from 73% to 88%) in pediatric emergency department visits [27] as well as a 48.4% reduction in admissions for acute myocardial infarction (AMI) [28]. This was accompanied by reports of delayed access to hospital care, with a parallel increase in fatality and complication rates [27,28]. These are concerning trends and they signal the need for community care to continue, even during a pandemic situation."

"In recognition of its potential to relieve hospital workloads and interrupt transmission during COVID-19, the South Korean Ministry of Health and Welfare temporarily lifted its ban on telemedicine, allowing numerous vulnerable patients to have access to continuing primary care from the safety of their homes [30]. The American Academy of Family Physicians (AAFP) also supports the expanded use of telemedicine as an important and efficient tool to help care for patients as the COVID-19 pandemic rages on [31]. As resources could be particularly scarce during a serious pandemic situation, timely psychological support could also take many forms [32]. Utilization rates of remote services, however, remain low globally consequent upon legal uncertainty, payment issues, technical challenges as well as patient and physician unfamiliarity [31,33]."

We hope this addresses your concern.

Reviewer 3 Report

Dear authors,
Thank you for your feedback and efforts to improve the description of the article.

Author Response

Thank you for the positive comments.